# Online Learning and Control of Dynamical Systems from Sensory Input

**Oumayma Bounou**[1], **Jean Ponce**[1,2], **and Justin Carpentier**[1]

[1]Inria and Département d'Informatique de l'Ecole Normale Supérieure, PSL Research University
[2]Center for Data Science, New York University

{oumayma.bounou, jean.ponce, justin.carpentier}@inria.fr

## Abstract

Identifying an effective model of a dynamical system from sensory data and using it for future state prediction and control is challenging. Recent data-driven algorithms based on Koopman theory are a promising approach to this problem, but they typically never update the model once it has been identified from a relatively small set of observations, thus making long-term prediction and control difficult for realistic systems, in robotics or fluid mechanics for example. This paper introduces a novel method for learning an embedding of the state space with linear dynamics from sensory data. Unlike previous approaches, the dynamics model can be updated online and thus easily applied to systems with non-linear dynamics in the original configuration space. The proposed approach is evaluated empirically on several classical dynamical systems and sensory modalities, with good performance on long-term prediction and control.

## 1 Introduction

### 1.1 Context and motivation

Providing complex machines such as robots with the capacity of learning effective models of their dynamics from their physical interactions with the world is a key enabling factor for deploying them outside of laboratories and factories. Analytical models of these interactions through state-based representations are traditionally derived from physical principles and used to solve challenging problems in various domains. They have been successfully deployed on heterogeneous systems such as robots or aircrafts/spacecrafts. Through the prism of control theory, these analytical models enable engineers to characterize and study the intrinsic dynamical properties of systems (contraction, stability, etc.), which are essential for system comprehension and for critical applications. In this context, optimal control theory [1] and its online counterpart known as model predictive control [2], exploit these models to plan and control complex dynamical systems (robots, rockets, etc.). Yet, the physical interactions of machines with the world are subject to complex natural phenomena, hardly describable by closed-form mathematical models.

By leveraging recent progress in machine learning, data-driven approaches [3, 4] have emerged as an effective way of learning advanced models of complex physical phenomena such as turbulence in fluid mechanics [3, 5, 6]. In particular, the Koopman formalism [7] is an attractive and versatile mathematical framework to learn and analyse complex dynamics. The overall objective of this formalism is to map the intrinsic state space of the system to a typically infinite-dimensional vector space where the system dynamics evolve linearly. However, exhibiting such a map remains challenging, especially in the context of instrumented systems such as robots, which evolve in the world by exploiting their

sensory input. The sensory inputs used to build a representation of the internal state of the system and the world may be of various nature: images and force, temperature values or encoder readings, etc. Importantly, to be effective for real-world applications such as the control of robotic systems, learned dynamical representations should also offer online update capacities, to account for new measurements as they come, which is essential to get long-term accurate prediction capacities in changing environments, and update the internal dynamics model accordingly.

## 1.2 Related work

**Learning dynamics from data.** Data-driven approaches [8, 9] based on the Koopman theory [7] and its approximation using the Dynamic Mode Decomposition (DMD) [3] have been used in the context of building dynamical models from measurements [6, 10, 11, 12, 13, 14, 15, 16, 17, 18]. A first group of approaches builds handcrafted functions to map from the measurements space to a latent space where the dynamics are forced to be linear [10, 11, 12, 13]. Such approaches are suitable when one specific system is being considered but cannot generalize to new systems, or to systems with varying physical properties. A second group of approaches learns a mapping from the measurements space to a latent space in the form of parametric functions. These approaches are more flexible, and showed promising results for both prediction [14, 15, 16, 17] and control [6, 18, 19]. [15], [16] and [17] learn an approximation of the DMD matrix in the form of a trainable linear layer. While this setting is satisfying when only a single dynamical system is considered, it becomes inappropriate to build more generic models since the same matrix can not be used to approximate the dynamics of different systems. To address this issue, [6, 14, 18] look for an approximation of the DMD matrix as a solution to a least-squares problem associated to a trajectory of a system. This enables identifying a linear model from a given trajectory of a system. In another line of work, [19] seek to identify a locally linear model of a system's dynamics in a learned representation space, through identifying a matrix to transition from one latent representation to the next one. However, their approach does not generalize to globally linear models.

**Learning and control of forced dynamics.** An extension of the DMD framework to actuated systems for model identification and control purposes has been introduced in [20], where the transition from one time step to the next is modeled by two matrices: the previous DMD matrix and a control matrix. Building upon this framework, [6, 16, 18] focused on learning an embedding for actuated systems. [6] and [16] parametrize the control matrix as a linear layer that is learned along with the embedding, while authors in [18] estimate the control matrix similarly to the DMD matrix (i.e.; from an input system trajectory representation), but assume they have access to the true state of the system, instead of only to measurements. [12, 13, 21, 22] also follow the same approach, although they do it on hand-crafted functions of either the measurements or the states, which they assume known in the case of [22]. In this work, we look for both the DMD and control matrices as solutions to a single least-squares problem including state representations and control. However, we assume the true state of the studied systems is unknown, and aim at leveraging sensory information to learn a linear representation of its dynamics. Contrary to the aforementioned approaches, we also introduce a proximal reformulation of the standard DMD approach based on a singular value decomposition (SVD) [23] to avoid introducing any extra hyperparameter of the rank truncation of the SVD that is required in the classical DMD setting. To account for changing dynamics, [22] estimate initial DMD and control matrices from an initial set of pairs of consecutive states of a given system, and perform active learning by updating their matrices with new executions of the system. In this work, we estimate matrices that are specific to a given trajectory, and update them with new measurements.

**Future video prediction.** The video prediction setting which is the focus of our paper has received a lot of attention in the past few years in computer vision, and various approaches for modeling the time evolution have been proposed. [24, 25, 26] do it through recurrent neural networks. Several approaches focused on predicting the next frame only conditioning on one previous frame [27, 28, 29, 30]. However, image data does not contain any velocity information. Indeed, if not explicitly modeled, this information can only be obtained when considering at least two consecutive frames, which is the approach we follow in our work. Some approaches [30, 31] model the multiplicity of future possible outcomes with variational approaches [32] however, in this work, we focus on the deterministic property of physical systems and only consider one possible future, like [33, 34, 35].

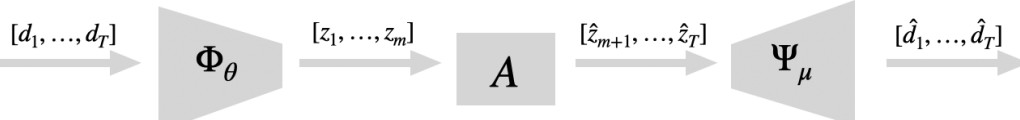

Figure 1: General overview of the proposed pipeline: the measurements $[d_1, \ldots, d_T]$ are first encoded with $\Phi_\theta$ to codes $[z_1, \ldots, z_T]$. Only the $m$ first codes are used to estimate the linear system dynamics $A$ arising from the least-squares minimization problem (3). Using this linear dynamics $A$ and the last code $z_m$, the last $T - m$ codes are predicted. The resulting reconstructed measurements $[\hat{d}_1, \ldots, \hat{d}_T]$ are obtained by decoding with $\Psi_\theta$ the embeddings of the actual measurements $[z_1, \ldots, z_m]$ and the predicted embeddings $[z_{m+1}, \ldots, z_T]$.

### 1.3 Contributions

We introduce an effective approach to learning a linear state representation directly from raw sensor input (e.g., images) in the form of an instance of a nonlinear latent embedding of a state space and a linear autoregressive model for that embedding [17]. Unlike classical approaches [6, 14, 15, 16, 17, 18, 36], once learned, our model of the system dynamics can be updated online to account for new measurements at very low cost. It also involves several key innovations: **(1)** Contrary to classical machine learning approaches to this problem [27, 31, 37], our approach is firmly grounded in (applied) Koopman theory [4, 7, 38] and, as demonstrated in Section 3.4, it readily generalizes from identifying the dynamics of a system to actually controlling it. **(2)** Conversely, contrary to existing algorithms based on Koopman theory and dynamic mode decomposition [3, 39, 40], we show that online updates of the linear dynamics model can efficiently be implemented in our framework, leading to improved prediction results (Sections 3.2 and 3.3). **(3)** We introduce several technical improvements that make the proposed approach practical, including a block-companion matrix representation of the linear autoregressive model enabling the use of multiple frames to drive video synthesis (Section 2), and the use of a robust, parameter-free proximal iterative refinement algorithm [41] for the least-squares estimation of this model which proves crucial in practice in the context of a deep learning library such as PyTorch. Our last main contribution is **(4)** an extensive experimental evaluation of the proposed approach on multiple problems from control theory (Section 3.4). Our experiments include video simulations of complex dynamical systems for which we can provide accurate long-term predictions and that we can effectively control.

## 2 Online learning of system representations

**Proposed approach:** We propose to learn the dynamics of a complex physical system from sensory inputs using a linear auto-regressive model for a nonlinear latent embedding of the corresponding state space [4, 6, 12]. Concretely, we consider a discrete-time dynamical system defined by some transition function $L : \mathcal{X} \to \mathcal{X}$ over some (arbitrary) state space $\mathcal{X}$. In our setting, $L$ is unknown, but its effect can be in fact (partially) observed through measurements (elements of $\mathbb{R}^p$, video frames in our case) acquired in successive states along trajectories of the dynamics. Our aim is to learn from such measurements (1) an implicit embedding $\varphi : \mathcal{X} \to \mathcal{Z}$ of the state space into some latent vector space $\mathcal{Z}$ using a parametric encoder $\Phi : \mathbb{R}^p \to \mathcal{Z}$ of the corresponding data, and (2) a linear model $A : \mathcal{Z} \to \mathcal{Z}$ of the dynamics in the latent space so that, for any $x$ in $\mathcal{X}$, $\varphi(Lx) = A\varphi(x)$. In this setting, $h = \Phi(d)$ is the feature $\varphi(x)$ associated with the (unknown) state $x$ in which $d$ has been measured. There is no reason to assume the dynamics $L$ to be linear (which may be meaningless anyway since $\mathcal{X}$ may not be a vector space, or more generally be endowed with a well-defined algebraic structure). However, we know from Koopman theory [7] that, when $\mathcal{Z}$ is taken to be the set of all real (or vector-valued) functions over $\mathcal{X}$, there exists a linear map $P : \mathcal{Z} \to \mathcal{Z}$, called the Perron-Frobenius operator [4, 42] such that $\varphi(Lx) = P\varphi(x)$ for any $x$ in $X$. Since $\mathcal{Z}$ is typically of infinite dimension in this case, $P$ does not normally admit a finite representation in the form of a matrix. In the practical case where the latent feature space $\mathcal{Z}$ is by design finite-dimensional, on the other hand, $P$ is not guaranteed to be linear but, given $m$ features $z_1, \ldots, z_m$ of dimension $n$, one can approximate its action using an $n \times n$ matrix $A$ minimizing the loss:

$$\|[z_2, \ldots, z_m] - A[z_1, \ldots, z_{m-1}]\|_F^2. \tag{1}$$

This approach is called Dynamic Mode Decomposition (DMD) [3] and, together with its extensions [20, 39, 43], it has been successfully used to study many dynamical systems. In our setting, the features are given by $z_i = \Phi(d_i)$ $(i = 1, \ldots, m)$ and correspond to $m$ successive measurements, gathered in our case from some video clip. DMD typically operates on raw measurement data instead of a latent representation thereof. Learning the embedding function $\Phi$ requires some sort of supervisory information, and we follow the popular encoder/decoder framework [44] by also estimating the parameters of a decoding function $\Psi : \mathcal{Z} \rightarrow \mathbb{R}^d$ such that $\Phi$ and $\Psi$ jointly minimize the regularized mean of the reconstruction error $\|d - \Psi \circ \Phi(d)\|_2^2$ over some sample of measurements. The full pipeline is composed of a learnable encoder/decoder module mapping from the measurement space to a latent state space, combined with a linear state representation which is learned from a sequence of embedded measurements (see Fig. 1). We now detail its different components.

**Measurement embedding.** The goal of the autoencoder is to learn an embedding of the measurements into a latent space representation of the state space. While input measurements may be high-dimensional objects (e.g. images), autoencoders are often capable of learning effective yet compact representations of the corresponding state space. Given a sequence of $T$ measurements $[d_{1:T}]$, we are looking for two parametric functions: an encoder $\Phi_\theta$ and a decoder $\Psi_\mu$ parametrized by $\theta \in \mathbb{R}^{n_\theta}$ and $\mu \in \mathbb{R}^{n_\mu}$, following the relation:

$$\Phi_\theta(d_t) = z_t \text{ and } \Psi_\mu(z_t) = \hat{d}_t \text{ for all t.} \tag{2}$$

**Dynamic mode decomposition** solves the least-squares problem associated with (1), which can be reformulated as:

$$\arg\min_A \frac{1}{2}\|Z_2 - AZ_1\|_F^2 \text{ where } Z_1 = [z_1, \ldots, z_{m-1}] \text{ and } Z_2 = [z_2, \ldots, z_m]. \tag{3}$$

A standard approach to solve this problem makes use of the singular value decomposition of the matrix $Z_1$ containing the collection of consecutive measurements, following the relation:

$$A^*(z_1, \ldots, z_m) = Z_2 Z_1^+, \tag{4}$$

where $Z_1^+$ is the pseudo-inverse of $Z_1$. As in practice $Z_1$ is a singular matrix, ones needs to carefully choose a given singular-value threshold, either to damp the solution or to cut the spectrum of $Z_1$. Both solutions only provide approximate solutions of the original least-squares problem (3). Additionally, the SVD is differentiable only when all the singular values are different [45], which may not be the case in general and could lead to unstable behavior when performing (stochastic) gradient descent throw computational layers which include SVDs. To overcome this issue, we propose to rely on an alternative approach called proximal method of multipliers [41]. Instead of solving a classical least-squares problem, we suggest solving an equality-constrained quadratic program of the form:

$$\min_{A \in \mathbb{R}^{n_z \times n_z}} \frac{1}{2}\|A\|_F^2 \text{ s.t. } Z_2 = AZ_1. \tag{5}$$

The proximal method of multipliers [41] augments the Lagrangian function associated to (5) with a proximal term over the dual variables associated to the constraint $Z_2 = AZ_1$, leading to:

$$L_\rho(A, \Lambda, \Lambda^-) = \frac{1}{2}\|A\|_F^2 + \sum_{t=1}^{T-1} \lambda_i^T (z_{t+1} - Az_t) - \frac{\rho}{2}\|\lambda_t - \lambda_t^-\|_2^2, \tag{6}$$

where $\rho$ is the smoothing parameter over the dual, $\lambda_t$ are the multipliers associated to the constraints $z_{t+1} - Az_t$, and $\lambda_t^-$ is an estimate of the multiplier $\lambda_t$. Such a saddle-point reformulation can be iteratively solved through the the Karush-Kuhn-Tucker (KKT) system of equations:

$$\begin{bmatrix} I_{n_z} & Z_1 \\ Z_1^T & -\rho I_{T-1} \end{bmatrix} \begin{bmatrix} A^k \\ \Lambda^k \end{bmatrix} = \begin{bmatrix} 0 \\ Z_2 - \rho\Lambda^{k-1} \end{bmatrix}, \tag{7}$$

where $I_{n_z}$ corresponds to the identity matrix of dimension $n_z$ and $\Lambda^k$ is the stack of multipliers associated to each constraint $z_{t+1} = Az_t$. (7) is solved by iterative refinement until convergence to the fixed point solution. Typically, for small values of $\rho$ (e.g. $10^{-8}$), less than a dozen of iterations are needed to converge to the optimal solution and the approch converges for any value of $\rho$. The system of equations (7) is always invertible thanks to the lower right block $-\rho I_{T-1}$ and has a condition number close to the one of $Z_1$. From a computational perspective, such a formulation can

be solved efficiently by exploiting the inherent sparsity of the 2x2 block matrix, using a Cholesky decomposition of the KKT matrix. Contrary to SVD, this approach can be easily differentiated by backward propagation of the gradient over the iterative process, which appears as an appealing feature for differentiable programming [46] in the context of deep learning.

**Online dynamic mode decomposition for long-term prediction.** Once built from a sequence of codes obtained from initial measurements, the DMD matrix $A$ can be used to predict the codes of future states given a known initial code. As classically done in [6, 40, 17], only the first $m$ measurements are used to estimate $A$, and future codes are predicted as:

$$\hat{z}_{m+t} = A^t z_m \text{ for } t \geq 1 \text{ s.t. } A^t \overset{\text{def}}{=} \underbrace{A \times \cdots \times A}_{t \text{ times}}. \tag{8}$$

In this framework, the matrix $A$ is never updated, thus never using potential new measurements. For long sequences, this might lead to inaccurate predictions as $A$ remains static and is only estimated from codes of the first $m$ measurements. While this is enough for simple systems (e.g, a pendulum), it becomes a limitation when more complex dynamical systems are considered (e.g, a cartpole). To overcome this issue, we propose to update $A$ when new measurements are acquired, making the learned representation adaptive. We account for this during training by artificially extending the initial sequence used to compute $A$ with additional latent codes of measurements. Re-starting prediction from codes of the new measurement should force the prediction error to diverge less quickly. We refer to the supplementary material for more details.

**Capturing dynamics effects.** Until now, we have considered one-step linear dynamics: each encoding $z_t$ only depends on the state $z_{t-1}$ linearly. However, most of the times, a measurement at a given time is only a partial observation of the state of the system and does not contain enough information to describe the full state of the system. For instance an image does not contain velocity information and at least two consecutive frames are needed to estimate it. To address this issue, we adopt a similar approach as proposed in [14] and consider a code $z_t$ which linearly depends on a history of $h$ previous codes $z_t^{:h} = [z_{t-h+1}, \ldots, z_t]$, such that:

$$z_{t+1} = A_1 z_{t-h+1} + A_2 z_{t-h+2} + \cdots + A_h z_t = A^{:h} z_t^{:h}, \tag{9}$$

where $A_i \in \mathbb{R}^{n_z \times n_z}$ and $A^{:h} = [A_1, \ldots, A_h]$. Like for Eq. (1), $A^{:h}$ can be obtained by solving an augmented linear least-squares problem, following the same proximal approach as before. Note that this formulation is equivalent to the one in Eq. (1), only augmented states $[z_t, \ldots, z_{t_{m-h+1}}]$ for $1 \leq t \leq m - h + 1$ would be considered, and the transition matrix would have a larger dimension of $hn \times hn$, depicting a block-companion structure of the form:

$$\tilde{A} = \begin{bmatrix} 0 & I & 0 & \ldots & 0 \\ 0 & 0 & I & 0 & \ldots \\ \ldots & \ldots & \ldots & \ldots & \ldots \\ 0 & \ldots & \ldots & 0 & I \\ A_1 & A_2 & \ldots & \ldots & A_h \end{bmatrix} \text{ s.t. } z_{t+1}^{:h} = \tilde{A} z_t^{:h}. \tag{10}$$

Such a scheme is classical when building models of dynamical systems from measurements and we refer to [47] for a comprehensive view.

**Learning forced dynamics for control.** Let us now consider an actuated system for which we have a sequence of measurements $d_{1:T}$. We assume the system was actuated with a sequence of control inputs $u_{1:T-1}$ that we have access to. The goal is to learn a representation space of the states of the system, from the measurements $[d_{1:T}]$ and the control inputs $[u_{1:T}]$ such that in this representation space, the evolution is linear on both states and control inputs. Following the extension of the DMD approach to control [20], we look for a latent space and matrices $A_1, \ldots, A_h$ and $B$ such that we have for all $t$:

$$z_{t+1} = A^{:h} z_t^{:h} + B u_t. \tag{11}$$

Unlike [6], we do not treat $B$ as a learned parameter. We also perform least-squares regression to find $A^{:h}$ and $B$ following the same approach as before. Once these matrices are identified, they can be exploited within the standard linear quadratic regulator setting [1] for control. This is possible because we make the assumption that $B$ is independent of $z$. In the case where this assumption does not hold, one can refer to the iterative linear quadratic regulator setting [48].

**Loss function.** To learn the parameters associated with our pipeline, we minimize the empirical risk of the $L_2$ reconstruction loss composed of two main terms: a first term associated to the autoencoder and a second term accounting for the desired linear prediction in the state space. The full loss is given by:

$$\mathcal{L}_{\theta,\mu}(\{d_{1:T}\}_{i=1,\ldots,N}) = \frac{1}{N}\sum_{i=1}^{N}\underbrace{\sum_{t=1}^{m}\|d_t^i - \Psi_\mu(\Phi_\theta(d_t^i))\|_2^2}_{\text{Auto-encoder loss}} + \underbrace{\sum_{t=m+1}^{T}\|d_t^i - \Psi_\mu(A_i^{t-m}\Phi_\theta(d_m^i))\|_2^2}_{\text{Prediction loss}}.$$

(12)

To simplify the notations, we do not account for the terms associated to the history ($A^{:h}$ and $z_t^{:h}$). In the online setting, predictions in the latent space following a measurement $d_j$ obtained at time $j > m$ are performed as $A^{t-j}\Phi_\theta(d_j)$ for $t > j$.

## 3 Results

We apply our learnable framework of systems dynamics from sensory input on challenging tasks. In particular, we address the tasks of future state prediction and control from raw video data, including both unactuated (unforced dynamics) and actuated (forced dynamics) systems.

### 3.1 Experimental setup

**Datasets.** We generate three black-and-white $64 \times 64$ video datasets of classical dynamical systems using the Pinocchio library [49] for the simulation and Panda3D-viewer [50] for the rendering: a simple pendulum, a double pendulum[1] and a cartpole. The cartpole we consider in our experiments is composed of a horizontal cart that is allowed one translation over an axis, to which a pole is attached and is allowed a rotation over an orthogonal axis. For each of these systems, we consider both unactuated and actuated versions, and generate training datasets composed of 4000 trajectories of 5 seconds ($T = 100$) in the first case, and 10 seconds ($T = 200$) in the second, with measurements every 50 ms. In the case of unactuated systems, we consider systems with varying physical parameters (masses, lengths) while in the case of actuated systems, we consider systems with the same physical parameters. In both cases, trajectories are generated with different initial conditions (angular positions and velocities). More details on the datasets' generation can be found in the supplementary material.

**Architecture.** The only learnable parameters in our model are those of the autoencoder. The encoder is made of 6 blocks of $3 \times 3$ convolutions followed by max-pooling, batch normalization and ReLu layers, except for the last block which does not have a ReLu layer. The decoder is a symmetric copy of the encoder. The dimension of the codes in latent space corresponds to the bottleneck size of the autoencoder, set to $n_z = 8$ in all our experiments. More details on the architecture can be found in the supplementary material. Each learning problem takes about 3 hours to solve on an Nvidia RTX6000 GPU.

**Evaluation.** The models of the unactuated systems are trained on $5\,\text{s}$ sequences and evaluated on $15\,\text{s}$ sequences, and those of actuated systems were trained on $10\,\text{s}$ sequences and evaluated on $20\,\text{s}$ sequences. Longer sequences are used for training actuated systems since in this case we seek to identify both dynamics ($A$) and control ($B$) matrices in latent space. For all systems, we have evaluated the prediction ability of trained models with the average RMSE loss over time. In Figures 3, 4, 5, the first vertical dashed line indicates the number of time steps used to estimate the dynamics matrix (and the control matrix in the case of forced dynamics), and the second vertical dashed line indicates the duration for which the models were trained.

### 3.2 Long-term prediction of unforced dynamics

Figure 3 shows the prediction error over an horizon of 15s for the pendulum and cartpole systems. For both systems, the error over the first time steps before the first vertical dashed line is the autoencoder error. Starting from the first vertical dashed line, the error is the prediction error. The red curve

---

[1]illustrated in the supplementary material.

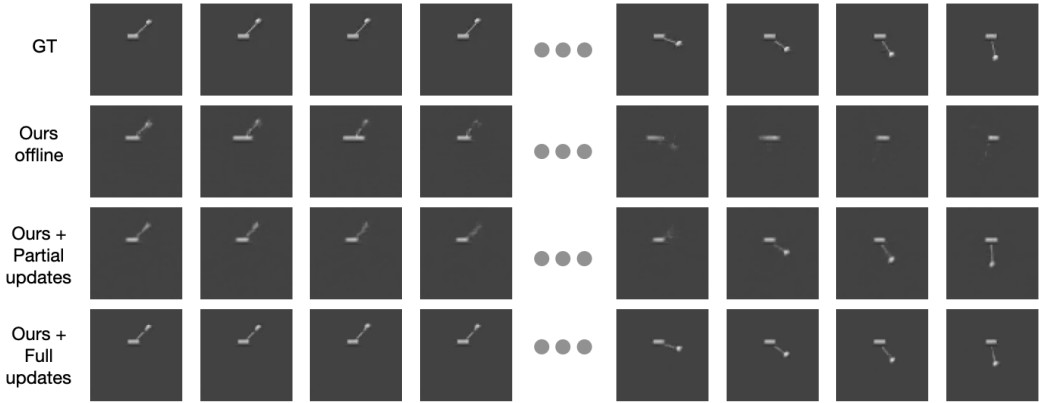

Figure 2: **Impact of online updates on the quality of the prediction for the cartpole.** The first row shows ground truth images. The second row shows predicted frames without updates. The third row shows predicted frames with our model trained with partial updates. The last row shows predicted frames with online updates performed at each new measurement.

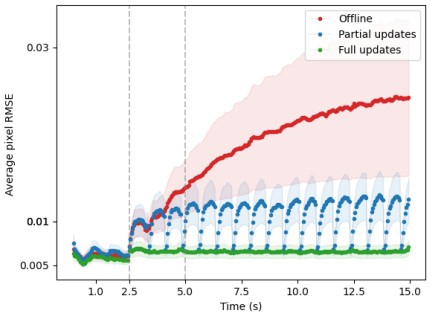 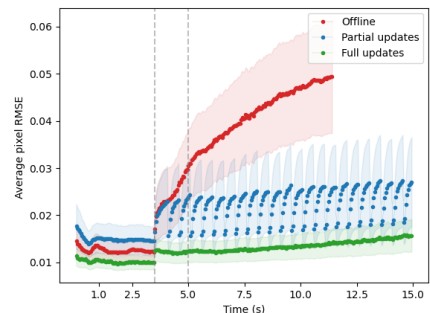

Figure 3: Average per-pixel RMSE loss over a 15s prediction horizon. See text for details. **Left:** pendulum. **Right:** cartpole.

corresponds to the case where the matrix $A$ is computed using a fixed number of time steps and never updated. The blue curve corresponds to the case where the matrix $A$ is first computed using a fixed number of time steps, then updated at fixed intervals (every $750\,$ms for the pendulum, and every $50\,$ms for the cartpole). The green curve corresponds to the case where the matrix $A$ is updated every 50ms. This case corresponds to predicting only one step ahead, which is why the error is low and constant over time. For both systems, the prediction error grows with time when the matrix $A$ is not updated, which demonstrates the necessity of online updates if one wants to perform accurate long-term prediction. The impact of the online updates on the quality of predicted frames of a cartpole can be seen in Fig. 2. The qualitative predictions are consistent with the evolution of the reconstruction error over time from Fig. 3 (right).

We compare our future prediction approach to the standard method of learning the matrix $A$ as a parameter of the model as in [17]. We train and evaluate both approaches on two different datasets of pendulums, the first one containing trajectories of the same pendulum (mass $m = 1$kg, length $l = 0.6$m) with different initial conditions, and the second one containing trajectories of different pendulums (mass $m = 1$kg, length $0.3 \leq l \leq 0.8$). Figure 4 highlights the fact that computing $A$ as the solution of a least-square problem (3) leads to more accurate predictions than learning with a fixed matrix $A$. In the supplementary material, we show that while the predictions are correct qualitatively on the first dataset even for a long horizon, they are wrong on the second dataset as not even the first predicted frame is correct. This is expected as a single matrix $A$ cannot be used to model the dynamics of systems with different physical parameters.

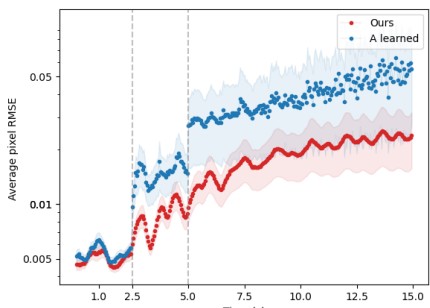 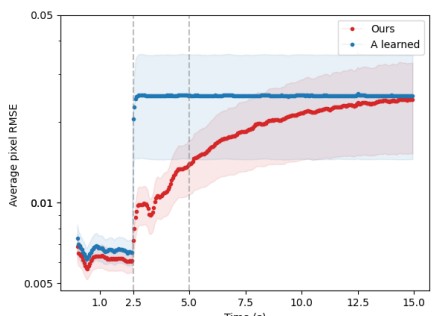

Figure 4: Average per-pixel RMSE loss over a 15s prediction horizon. **Left:** pendulum with length 0.6 m. **Right:** pendulums with lengths varying from 0.3 to 0.8 m.

## 3.3 Long-term prediction of forced dynamics

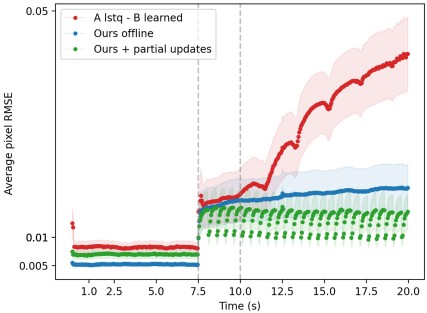 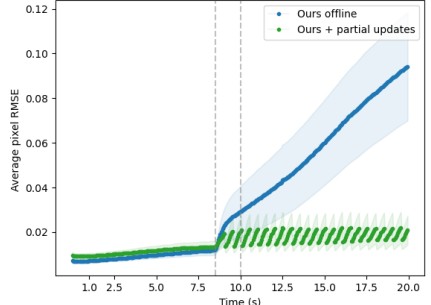

Figure 5: Average per-pixel RMSE loss over a 20s prediction horizon. **Left:** actuated pendulum. **Right:** actuated cartpole.

We consider the case of controlled pendulum and cartpole systems. Figure 5 shows the prediction error over a horizon of 20s of actuated pendulum and cartpole. The importance of (partial) online updates is demonstrated as without it the prediction error grows quickly over time.

We have compared our approach to the approach presented in [6], where the dynamics matrix $A$ is the result of a least-square problem similar to ours, while $B$ is a linear layer learned as a trainable parameter along with the autoencoder parameters. We adapt their approach to our setting in order to **(1)** include time-delay embeddings because our measurements are images, and **(2)** use an autoencoder tailored to our datasets, for fairer comparison. Even for offline prediction, our approach outperforms our implementation of [6] in the case of the actuated pendulum (Fig. 5, left), which validates the need for a learned specific control matrix $B$ per trajectory. Having a constant $B$ would imply that a given sequence of control inputs has the same effect on the state of the system no matter when along in the trajectory it is applied, presupposing a stationary regime of the states of the system, which is not the case for most real physical systems. This is all the more validated with the cartpole example in Fig. 5 (right), for which the approach with a static matrix $B$ is unable to make accurate predictions in the 2.5s horizon, because of the chaotic regime of the system. More details on shorter-term prediction horizons of this approach can be found in the supplementary material.

## 3.4 Control from video inputs

We demonstrate the effectiveness of our approach on the control of simulated physical systems directly from visual inputs. We consider here the case of controlling a forced system such as a

pendulum. Such a system is highly non-linear and does not exhibit any linear state representation. Using our pipeline, we learn a linear-state representation $(A, B)$ which can then be plugged into classic linear quadratic control settings [1] similarly to [51, 12, 18, 6].

Let us assume the system is at some initial configuration $d_1$ in the measurements space, and our goal is to drive it to a target configuration $d_f$ given in the image space at time $T_c$. We are looking for a sequence of control inputs $[u_1, \ldots, u_{T_c}]$ that minimizes:

$$\min_u \sum_{t=2}^{T_c} (z_t - z_f)^T Q (z_t - z_f) + \sum_{t=1}^{T_c-1} u_t^T R u_t \quad \text{s.t.} \quad z_{t+1} = A z_t + B u_t \text{ and } z_1 = \Phi_\theta(d_1), \quad (13)$$

where $Q$ and $R$ are the standard symmetric positive definite matrices associated to the LQR setting.

Figure 6 shows the trajectory obtained for driving the pendulum from the initial positions (red) to a target configuration (green). As our representation is with delay embeddings with an history of 2 frames, we need to constrain the QP problems on two initial conditions. It is important to notice that we were not able to run this experiment with a single image history. Indeed, a pendulum and other mechanical systems are subject to gravity, which is a second-order information, which cannot be retrieved from a single image.

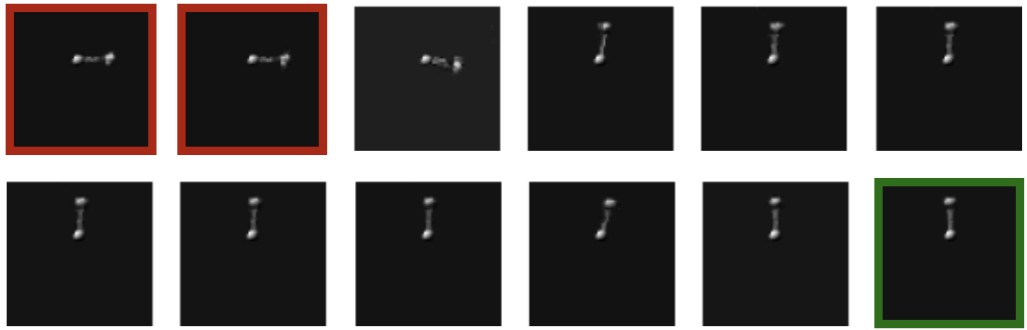

Figure 6: **Illustration of pendulum control.** Starting from two consecutive configurations (in red), control inputs are applied to the pendulum to force it to be in an inverted position in a horizon of $0.5$ s. Final position is the target configuration (in green).

## 3.5 Limitations

In all the experiments, we use an embedding of the state of relatively small dimension $n_z = 8$ compared to the configuration space dimension of the studied dynamical systems. Yet, while we manually choose this value, the issue of choosing the correct dimension of linear state space representations remains open. Additionally, all our experiments were done on simulated and low-dimensional systems without considering noise on the sensory inputs. A next step to our approach would be to include real systems or measurements of real systems of higher dimension (e.g, real video datasets). Finally, our approach is motivated by the Koopman operator theory [7], through the DMD approximation [3]. While this approximation works in practice, no theoretical bound of this approximation that we are aware of has been established yet, and we believe that finding one remains an open problem. Some previous work ( [43], [52]) discussed this issue from a spectral point of view by either showing that DMD (or EDMD) modes are a good approximation of the true Koopman eigenfunctions for specific problems when these are known analytically, or that they provide a correct parametrization of the system. Again, as far as we know, assessing the accuracy of the DMD/EDMD approximation to the Koopman operator for nonlinear dynamics remains an open problem.

## 4 Conclusion

We have introduced a trainable framework for learning an embedding of the state space of physical systems into a latent space where the dynamics are linear, directly from raw sensor inputs (images). Contrary to previous works, our computational framework allows learning system state representations

which are adapted to online updates, leading to more accurate predictions than offline models. This feature is also essential for online control of complex instrumented systems such as robots, where a compact state representation needs to be updated to account for dynamical changes in the environment. Additionally, our framework exhibits long-term prediction capabilities, an essential feature for planning and control. It is also able to learn complex embeddings for dynamical systems with varying parameters, which is an appealing feature for systems which operate in changing environmental conditions. We have applied it to future state prediction and control from input videos on several challenging dynamical systems. As future work, we plan to extend our approach to operate on a heterogeneous set of sensor inputs and to apply it on real robotic systems for solving fine manipulation tasks by directly exploiting vision and tactile feedback, similar to the experimental setting developed in [37]. We plan to extend our approach to 3D examples by accounting for multi-view measurements, in a similar spirit to what authors in [53] proposed.

**Broader impact.** This work is mostly a theoretical contribution to the online learning of complex dynamics from sensory input and its potential positive or negative impacts are similar to the ones of the field. Applications such as the learning of complex dynamics for machines interacting with the world or humans, among them manipulation or locomotion tasks in robotics, may lead to questionable use in surveillance or military contexts. Yet, we believe that developing generic approaches to learning complex dynamics with structured and interpretable components is part of a more general trend towards more structured learning algorithms. This could results in more explainable and efficient algorithms, a central topic for the ethical and ecological issues of AI.

# 5   Acknowledgements

We warmly thank Armand Jordana for fruitful discussions. This work was supported in part by the HPC resources from GENCI-IDRIS (Grand 2020-AD011011263R1), the Inria/NYU collaboration, the Louis Vuitton/ENS chair on artificial intelligence and the French government under management of Agence Nationale de la Recherche as part of the "Investissements d'avenir" program, reference ANR19-P3IA0001 (PRAIRIE 3IA Institute).

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
