# Online Learning and Control of Complex Dynamical Systems from Sensory Input - Supplementary Material

**Oumayma Bounou**[1], **Jean Ponce**[1, 2]**, and Justin Carpentier**[1]

[1]Inria and Département d'Informatique de l'Ecole Normale Supérieure, PSL Research University
[2]Center for Data Science, New York University

## 1 Additional details on the experiments

### 1.1 Model architecture

The only learnable parameters (117,963 in all) in our model are those of the autoencoder. The encoder is made of 6 blocks of $3 \times 3$ convolutions with 16, 32, 64, 64, 32 and 8 channels followed by max-pooling, batch normalization and ReLu layers, except for the last block which does not have a ReLu layer. The decoder is a symmetric copy of the encoder. As our images are $64 \times 64$, the last convolutional block yields a feature map with 8 channels and $1 \times 1$ spatial dimension, which is reshaped into an $8 \times 1$ vector. The latent code we consider is thus directly the output of the convolutional encoder. Contrary to [1], we do not follow our encoder by fully-connected layers to obtain a compact code since the output of the convolutional encoder is alreay quite compact.
Models without updates take 2.5 hours to train on a Tesla V100-SXM2 GPU, and models with updates take 4 hours to train. Models including control take longer to train (4 hours without updates and 6 hours with partial online updates) since the video sequences considered are longer. All models are trained for 200 epochs with a batch size of 16 and a learning rate of $10^{-3}$ which is divided by 2 every 20 epochs.

### 1.2 Experiments on pendulum systems

#### 1.2.1 Datasets generation

We have generated video datasets of cartpole and pendulum systems to which control inputs are applied. The length of the generated videos is $10\,\text{s}$ and points of the system are generated every $5\,\text{ms}$, which is also the frequency at which controls are applied. Measurements (i.e, images) are taken every $50\,\text{ms}$. In the following, time steps will refer to measurements time steps (every $50\,\text{ms}$). To obtain videos of actuated systems, we have generated a set of reference trajectories and velocities to be followed by the systems. For simplicity, we specifiy the angle between the pole and the vertical, as well as its temporal derivative, and also control these two quantities. Thus in the case of the cartpole, only the pole is actuated, the translation of the cart remains free. Each reference trajectory is the sum of three sinusoidal signals with different frequencies. Starting from a random initial configuration, the system (pendulum or cartpole) receives a control input every $5\,\text{ms}$ to match the target trajectory. The reference trajectories are of the shape:

$$\begin{cases} q_{ref}(t) = \sum_{i=1}^{3} q_{0,i} \sin(\omega_i t + \varphi_i) \\ v_{ref}(t) = \sum_{i=1}^{3} q_{0,i} \omega_i \cos(\omega_i t + \varphi_i), \end{cases} \tag{1}$$

where $q_{0,i}$ is the angular amplitude of the reference trajectory for the pole. In our experiments, $q_{0,i}$ was uniformly sampled between 0 and $\frac{\pi}{3}$ radians for the pendulum, and between 0 and $\frac{2\pi}{3}$ radians for the cartpole. We take $\omega_i = 2\pi f_i$ where $f_i$ is uniformly sampled between 0 and 0.1 Hz for the

35th Conference on Neural Information Processing Systems (NeurIPS 2021).

cartpole system and between $0$ and $0.3$ Hz for the pendulum system. We take $\varphi_i$ between $0$ and $2\pi$ radians for both systems.

Having generated these reference trajectories, the control input to apply to the systems every $5$ ms is determined as:

$$u(t) = -K_p(q(t) - q_{ref}(t)) - K_d(v(t) - v_{ref}(t)), \qquad (2)$$

with $K_p = 100$ and $K_d = 10$.

Sinusoidal inputs and sums of sinusoidal inputs are commonly used to excite systems as they facilitate system identification [2]. This is why we choose to use reference trajectories in the form of a sum of sinusoids, since the controls we apply to the system are proportional to these trajectories, as can be seen in Eq. (2).

### 1.2.2 More results

**Prediction quality.** In all the following figures, the left block corresponds to the first predicted frames after those used to compute the matrix $A$, and the right block corresponds to predicted frames after a horizon of 20 or 30 time steps. Figure 1 shows that for a simple system such as a pendulum with low amplitude oscillations (between $\frac{\pi}{2}$ and $\frac{3\pi}{2}$ radians), our offline model without any update is sufficient to predict future frames correctly. However, for more complex systems such as the pendulum with high amplitude oscillations (up to $2\pi$ radians) (Fig. 4, second row) or the double pendulum (Fig. 5, second row), our offline model yields blurry predictions, that can be corrected by performing online updates (even if they are not performed at every new measurement, but only every 15 measurements) (Figures 4 and 5, third row). Finally, for both systems, performing online updates at each new measurement yields visually perfect predictions at all time steps. (Figures 4 and 5, fourth row). The quality of the predictions for the double pendulum and the pendulum with high oscillations amplitude is consistent with the time evolution of the RMSE loss for these systems (Fig. 6), whose values are most likely over optimistic since most pixel values are 0 in our datasets.

Figures 2 and 3 compare the quality of the predictions of our model to the quality of prediction of the baseline where the matrix $A$ is learned as an additional parameter, and is thus constant over all the dataset. In this case, the matrix $A$ is not computed using codes of past frames, it is instead learned, along with the parameters of the autoencoder. There is thus one single matrix $A$ that is used for the prediction of future frames of different trajectories. Figure 2 shows that when models are trained on a dataset of a single pendulum with different trajectories (different initial conditions: initial position and velocity), the baseline gives good short-horizon predictions (left block) but poor predictions for longer horizons (right block). Our model does not exhibit such limitations. Figure 3 shows how the baseline model is unable to predict future frames correctly, for even a single step in the future (first frame of the left block), when it is trained on a dataset with multiple pendulums. This is not surprising as a single matrix $A$ can not account for the dynamics of several different systems.



Figure 1: **Prediction for the pendulum with low oscillations amplitude (between $\frac{\pi}{2}$ and $\frac{3\pi}{2}$ radians) on a dataset of pendulums of length varying between $0.3$ m and $0.8$ m.** The first row shows ground truth (GT) images. The second row shows predicted frames with our model without updates. In the case of this simple system, our model without updates is enough.

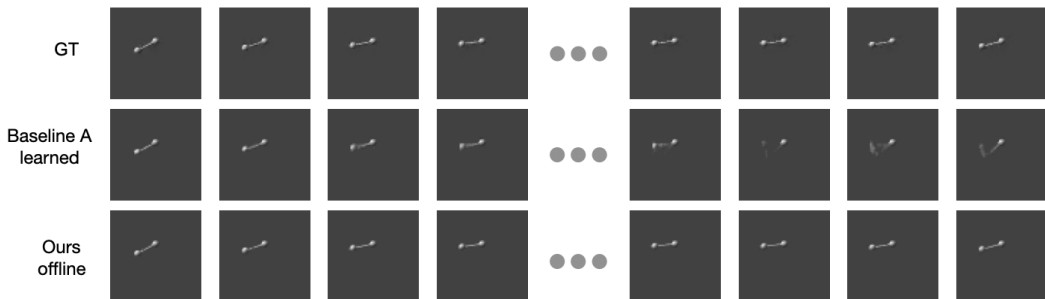

Figure 2: **Prediction for the pendulum with low oscillations amplitude (between $\frac{\pi}{2}$ and $\frac{3\pi}{2}$ radians) and length** $0.6$ **m.** The first row shows ground truth (GT) images. The second row shows predicted frames without updates with the baseline model where the matrix $A$ is a learned parameter. The third row is our model without updates.



Figure 3: **Prediction for the pendulum with low oscillations amplitude (between $\frac{\pi}{2}$ and $\frac{3\pi}{2}$ radians) with lengths varying from** $0.3$ **to** $0.8$ **m.** The first row shows ground truth (GT) images. The second row shows predicted frames with the baseline model where the matrix $A$ is a learned parameter.

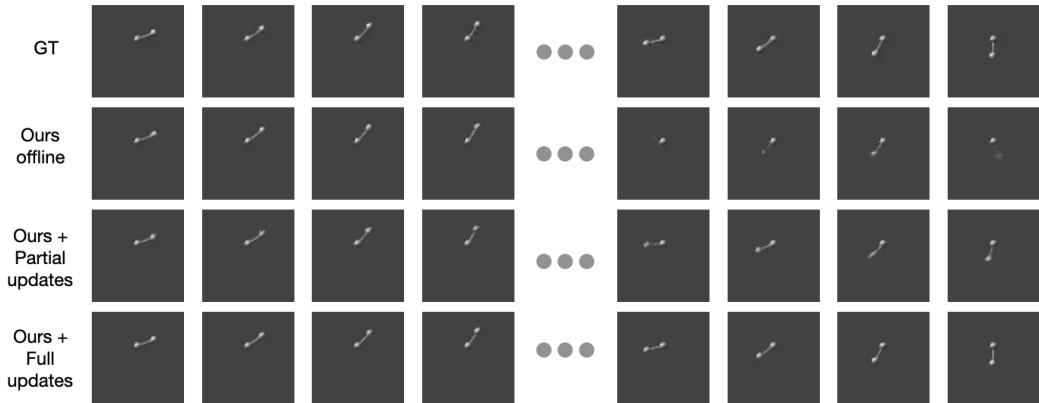

Figure 4: **Impact of online updates on the quality of the prediction for the pendulum with high oscillations amplitude (between** $0$ **and** $2\pi$ **radians) on a dataset of pendulums of length varying between** $0.3$ **m and** $0.8$ **m.** The first row shows ground truth (GT) images. The second row shows predicted frames without updates. The third row shows predicted frames with our model trained with partial online updates (every 15 measurements). The last row shows predicted frames with online updates performed at each new measurement.

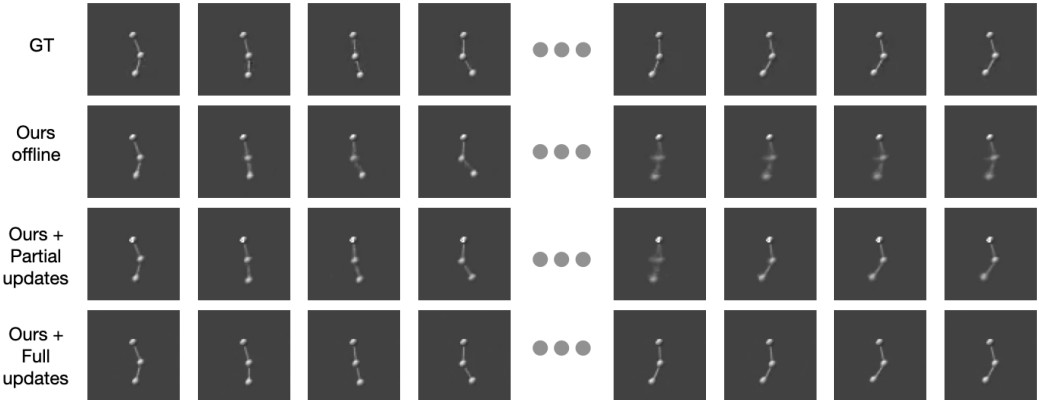

Figure 5: **Impact of online updates on the quality of the prediction for the double pendulum.** The first row shows ground truth (GT) images. The second row shows predicted frames without updates. The third row shows predicted frames with our model trained with partial online updates (every 15 measurements). The last row shows predicted frames with online updates performed at each new measurement.

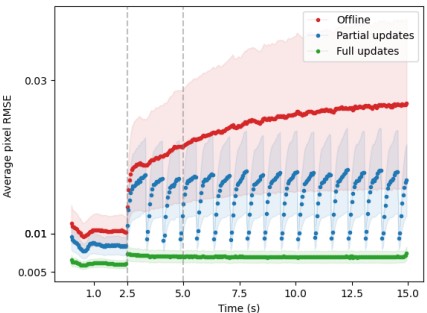 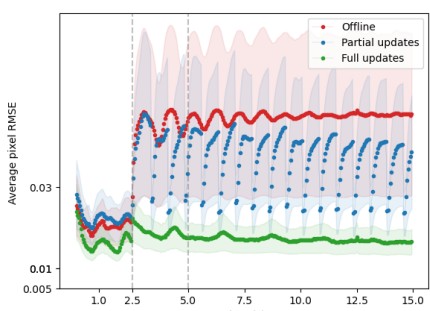

Figure 6: Average per-pixel RMSE loss over a 15s prediction horizon. **Left:** Pendulum with high amplitude oscillations (between 0 and $2\pi$ radians). **Right:** Double pendulum with a first pole oscillating between $\frac{\pi}{2}$ and $\frac{3\pi}{2}$ radians.

**Control.**
Figure 7 shows the trajectory obtained when driving the cartpole from an initial state specified by two consecutive frames (red) to a position where the pole is inverted. [1] Solving the QP problem of Eq. (13) of the main submission returns a sequence of controls $[u_1, \ldots, u_{10}]$ that are applied starting from $z_0$ and $z_1$, the embeddings of the two first frames (red), such that for $0 \leq t \leq 11$:

$$z_{t+2} = A_1 z_t + A_2 z_{t+1} + B u_{t+1} \qquad (3)$$

where $A_1$, $A_2$ are blocks of the matrix $\tilde{A}$ described in Eq. (10) of the main submission, in the case where $h = 2$. The frames are obtained by decoding the sequence $[z_0, \ldots, z_{11}]$.

---

[1] As described in the main submission, our prediction model uses not one but at least two codes in the latent space to predict the next one, which is why, for the control task, two initial frames are considered and used to constraint the QP problem described in Eq. (13) of the main submission.

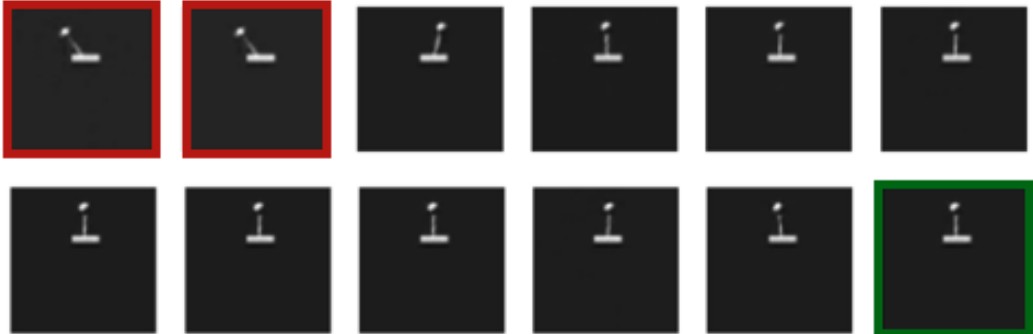

Figure 7: **Illustration of cartpole control.** Starting from an initial state specified by two consecutive frames (in red), we estimate and apply the controls necessary to guide the pole to an inverted position in a horizon of $0.5\,\mathrm{s}$. The last frame shows the final position of the cartpole after all controls were applied (in green).

## 1.3 Experiments on fluids

We extended our approach to the study of a fluid flowing past a cylinder following [1], through the study of four of its physical quantities: density, x-momentum, y-momentum and energy.

### 1.3.1 Dataset generation

We followed the dataset generation protocol described in [1]: the solver PyFR [3]is used to solve the Navier-Stokes equation [4] for each one of the four quantities mentioned in the previous paragraph, with a discretization time step of 0.1 ms. The solutions are then formatted into 4-channels image-like inputs of size $128 \times 256$ (one channel per physical quantity), and one image is kept every 150 ms (every 1500 steps of the solver) for each of the four quantities. The simulation is run in two different settings: unforced and forced dynamics. In the unforced dynamics setting, the simulation is run during 636 seconds (which corresponds to a trajectory of 4328 frames) with no velocity being prescribed to the cylinder. In the forced dynamics setting, the simulation is run during 750 seconds, (which corresponds to a trajectory of 5000 frames) with a velocity being prescribed to the cylinder. The obtained trajectories are then split into respectively 1200 and 1600 overlapping sequences, both with a duration of 4.8 s.

### 1.3.2 Experimental protocol

We trained our model during 1000 epochs on 1200 32 frame-long (4.8 s) sequences in the case of unforced dynamics, and on 1600 32 frame-long (4.8 s) sequences in the case of forced dynamics. In this 32 frame-long sequence, encodings of the 16 first frames (2.4 s) were used to estimate the dynamics matrix $A$ (and the control matrix $B$ in the case of forced dynamics), and the 16 (2.4 s) following frames were predicted. We used a batch-size of 16 and a latent dimension $n_z = 8$ in the case of unforced dynamics, and $n_z = 32$ in the case of forced dynamics. We set our initial learning rate to $1e^{-3}$, and divided it by 2 every 100 epochs.

### 1.3.3 Results

**Prediction.** We evaluated our model on 100 frame-long (15 s) sequences. Figure 8 shows the average L1 loss over time over 120 test trajectories using the three variations of our approach we detailed in the main paper. Even though we see that variations of our model that include updates (at every time step starting the 16th time step in green, or at a single time step, the 40th, in blue) have lower error values that do not grow over time compared to our offline variation (where no update of the model is performed), the error values for all three variations remain very low and are invisible to the naked eye, as can be seen in Fig. 9.

In this work, we followed the experimental protocol described in [1] for comparison, however, we believe that future work should consider multiple trajectories from different fluids (with different

physical parameters) instead of only one unique trajectory of one fluid, and that the sequences used for training should not overlap.

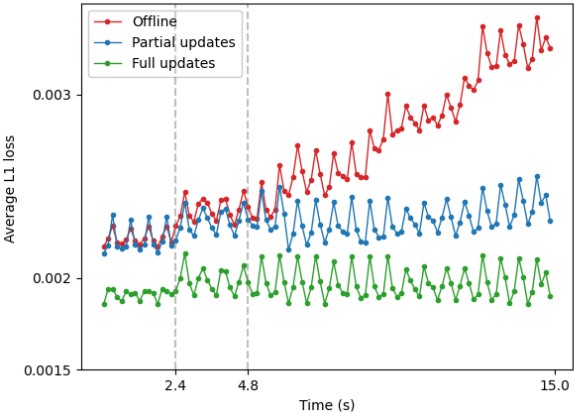

Figure 8: Average L1 loss on all four quantities of the fluid system over a 15 s prediction horizon.

**Control.**
Figure 10 shows the trajectory of the x-momentum of the studied fluid obtained when applying a sequence of controls to stabilize the fluid flow. The controls are a solution to the QP problem defined in equation (13) of the main submission where $z_1$ corresponds to an initial representation of the fluid, and $z_f$ corresponds to a representation of the fluid where it is stabilized (i.e.; when its flow is laminar). Note that each code $z_t$ is built by encoding the 4 physical quantities mentioned above at time $t$ using the learned encoder, and that the resulting controlled sequence in Fig. 10 only shows one quantity (the x-momentum).

## 1.4 Training details

During training, we seek to minimize the loss defined in equation (12) of the main paper. For ease of reading, equation (12) only accounts for the case of unforced dynamics (i.e.; where the studied systems are not actuated, thus when we are only looking for the dynamics matrix $A$). In the case of actuated systems, an additional term is added to this loss such that it becomes:

$$\mathcal{L}_{\theta,\mu}(\{d_{1:T}\}_{i=1,\ldots,N}) = \frac{1}{N}\sum_{i=1}^{N}\underbrace{\sum_{t=1}^{m}\|d_t^i - \Psi_\mu(\Phi_\theta(d_t^i))\|_2^2}_{\text{Auto-encoder loss}} + \underbrace{\sum_{t=m+1}^{T}\|d_t^i - \Psi_\mu(A_i^{t-m}\Phi_\theta(d_m^i) + B_i u_t^i)\|_2^2}_{\text{Prediction loss}}.$$

(4)

At each optimization step, $A_i$ and $B_i$ are estimated using equation (5) from the main paper for each trajectory $i$ $[d_1^i,\ldots,d_T^i]$. In fact, they are estimated from the first $m$ codes of $[d_t^i]_t$ (obtained with the encoder $\Phi_\theta$), then used to predict future codes through the relation $z_{t+1}^i = A_i z_t^i + B_i u_t^i$. The sequence $[z_t^i]_t$ is then decoded using the decoder $\Psi_\mu$. The parameters $(\theta, \mu)$ of $\Phi_\theta$ and $\Psi_\mu$ are then updated. In practice, we see that the term $\sum_{i,t}\|z_{t+1}^i - (A_i z_t^i + B_i u_t^i)\|_2^2$ decreases during training without being explicitly minimized, as can be seen in Fig. 11.

## 2 Online updates

The estimation of the matrix $A$ requires inverting the matrix:

$$M = \begin{bmatrix} I_{n_z} & Z_1 \\ Z_1^T & -\rho I_{T-1} \end{bmatrix}.$$

(5)

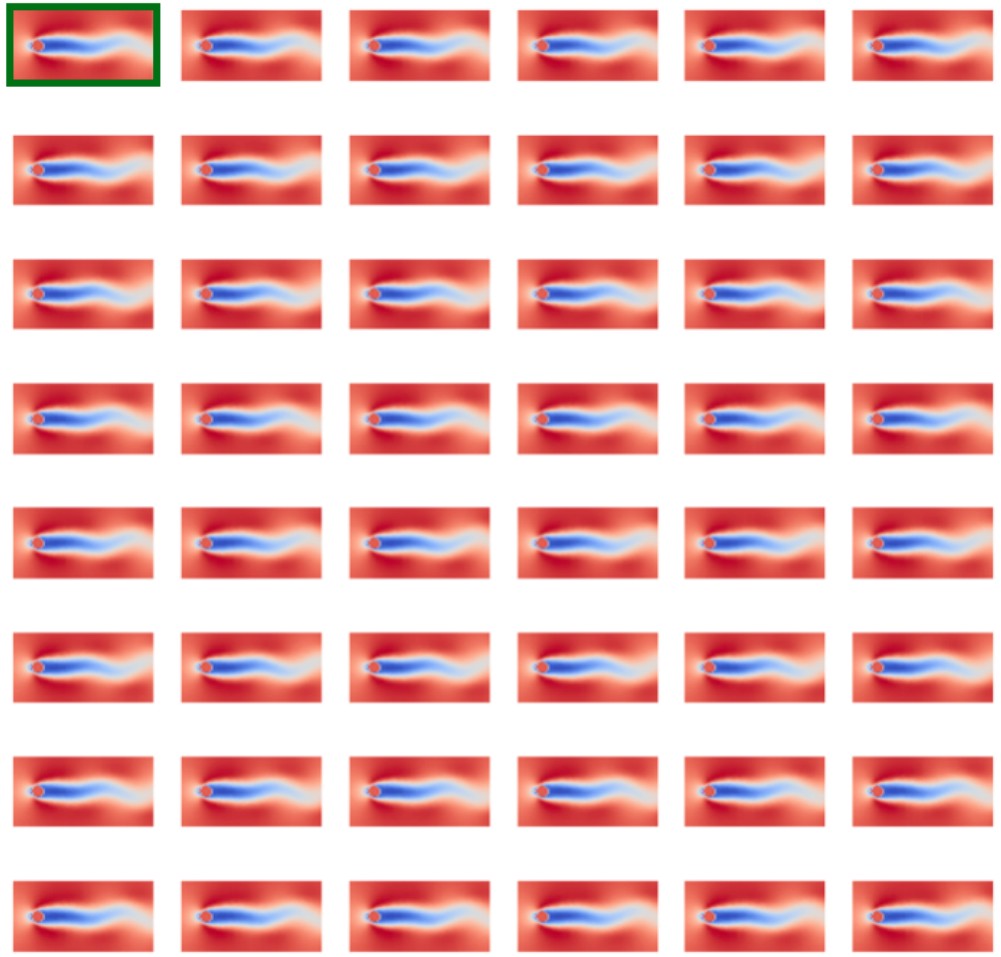

Figure 9: **Prediction of the x-momentum**. The top-left frame is the last frame of the 16 frame-long sequence that was used to build the dynamics matrix $A$. All the following frames are predicted.

This can be efficiently performed through a Cholesky decomposition of the form $LDL^t$ because $M$ is the KKT matrix associated to a saddle point problem, and has positive definite and negative definite blocks. In the case where our model is updated online, $M$ must be recomputed at every update. We can avoid recomputing it from scratch by performing rank-1 updates of its Cholesky decomposition when new measurements are considered (which would correspond to adding one column to $Z_1$ and one row to $Z_1^T$).

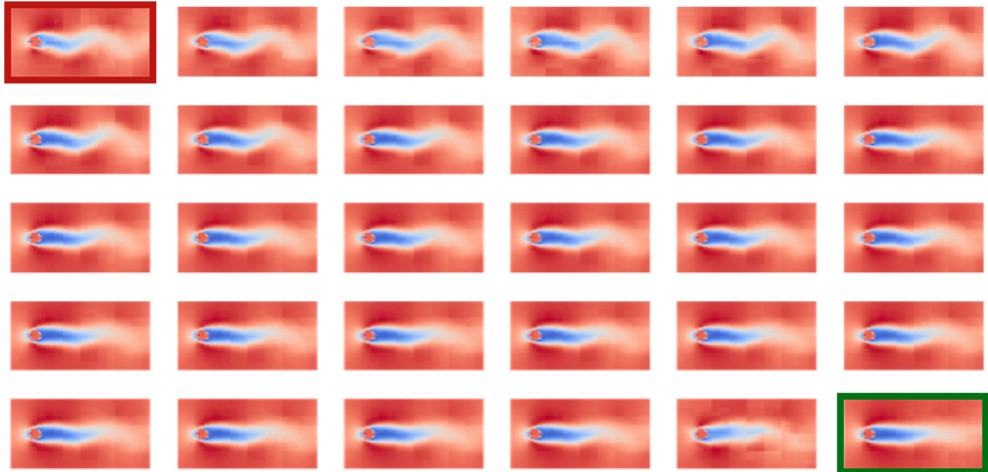

Figure 10: **Illustration of fluid control**. Starting from an initial configuration of the fluid at a given time step (top-left), we estimate and apply (in the learned latent space) the controls necessary to stabilize it (bottom-right). The quantity shown here is the x-momentum.

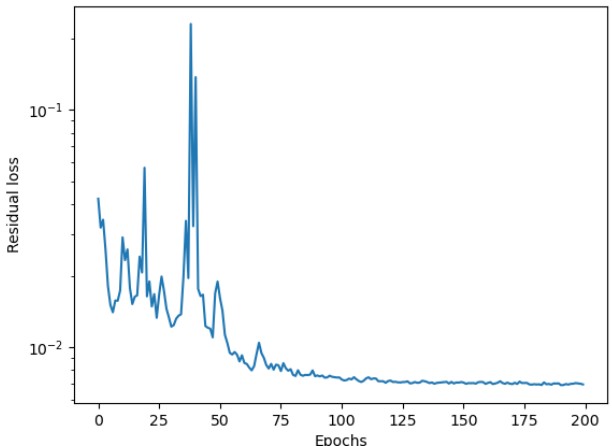

Figure 11: **Residual loss**. Evolution of $\sum_{i,t} \|z_{t+1}^i - (A_i z_t^i + B_i u_t^i)\|_2^2$ during training.