# OpenReview forum: "Online Learning and Control of Complex Dynamical Systems from Sensory Input"
_NeurIPS.cc/2021/Conference — NeurIPS 2021 Poster_

### Official Review · Reviewer_3pLg · 2021-07-15

**Rating:** 6
**Confidence:** 3

**Summary:**

This paper discusses the problem of controlling a dynamic system based on raw sensory inputs. Their method uses an encoder/decoder to map the sensor information (video frames in this paper) to a learned embedded space. Next, it proposes to build a linear function that evolves the dynamic system’s states in the embedded space. This paper evaluates the performance of the proposed pipeline on three classic control problems: cart-pole, single pendulum, and double pendulum.

**Limitations And Societal Impact:**

This paper did a good job of discussing its limitations in its low-dimensional, simulation-only experiments. I agree with these limitations, but instead of leaving them as future work, I would suggest trying to solve them in this submission so that it could be a competitive paper.

I do not see major societal and ethical issues from the proposed method, and I appreciate that the authors explicitly discuss them in the paper.


**Main Review:**

While the proposed pipeline seems reasonable, the method is evaluated on three classical control problems only. In my opinion, a NeurIPS paper needs more competitive examples.
- It is a bit too much to call cart-pole/(double) pendulum “complex dynamical systems” in the title/abstract/introduction. Please consider toning down the writing in title/abstract/introduction.
- The discussion on Eqn. (3) and (5) would be clearer if some context about the dimension of n and m could be given. As far as I understand, Eqn. (3) and (5) are two quite different optimization problems that should be used in different scenarios: if AZ_1 = Z_2 is an under-determined system (e.g., n >> m), then Eqn. (5) is preferred. On the other hand, if AZ_1 = Z_2 is a overdetermined system (n << m), one should stick to Eqn. (3) because the equality condition AZ_1 = Z_2 in Eqn. (5) may not be feasible at all. Which case is this paper faced with?
- From what I understand, all three examples (cart-pole, pendulum, and double pendulum) deal with 2D dynamics only because their motions are constrained on 2D planes, although they can be rendered as if they are 3D scenes. I would like to see a 3D example (e.g., quadrotor, ant, cheetah, or humanoid robots) in simulation, and more importantly, I am curious to see if the pipeline from video frames to embedded space still works for a 3D task. I would guess this would create some additional challenges due to the ambiguities in understanding 3D scenes from 2D images.


**Time Spent Reviewing:**

3.5

---

> ### Author Response · Authors · 2021-08-10
> **Response to reviewer 3pLg**
>
> Thank you for your comments and your helpful suggestions. We will try to address each one of your remarks in the following:
>
> > It is a bit too much to call cart-pole/(double) pendulum “complex dynamical systems” in the title/abstract/introduction. Please consider toning down the writing in title/abstract/introduction.
> - We will tone down the writing in the abstract/introduction when referring to the cartpole and (double) pendulum systems.
> We have recently extended our approach to computational fluid dynamics (CFD) data. We refer the reviewer to the common answer for more details.
>
> ___
> > The discussion on Eqn. (3) and (5) would be clearer if some context about the dimension of n and m could be given. As far as I understand, Eqn. (3) and (5) are two quite different optimization problems that should be used in different scenarios: if AZ_1 = Z_2 is an under-determined system (e.g., n >> m), then Eqn. (5) is preferred. On the other hand, if AZ_1 = Z_2 is a overdetermined system (n << m), one should stick to Eqn. (3) because the equality condition AZ_1 = Z_2 in Eqn. (5) may not be feasible at all. Which case is this paper faced with?
> - In our experiments, the embedding dimension $n$ was set to 8 for all three considered systems in the submission (and also in CFD data), while $m$ varies between 50 and 150 for the considered systems in the submission (and between 32 and 64 for CFD data). Chiche et al. showed in [https://hal.archives-ouvertes.fr/hal-01057577/] that problem (3) is included in problem (5). They notably showed that when the constraint $Z_2=AZ_1$ is feasible (under-determined), solving problem (5) returns a minimal norm solution, while when problem (5) is not feasible, the augmented Lagrangian strategy returns the least-square solution of $Z_2=AZ_1$ (over-determined). We will include this distinction in our revised paper.
> ___
> > From what I understand, all three examples (cart-pole, pendulum, and double pendulum) deal with 2D dynamics only because their motions are constrained on 2D planes, although they can be rendered as if they are 3D scenes. I would like to see a 3D example (e.g., quadrotor, ant, cheetah, or humanoid robots) in simulation, and more importantly, I am curious to see if the pipeline from video frames to embedded space still works for a 3D task. I would guess this would create some additional challenges due to the ambiguities in understanding 3D scenes from 2D images.
> -  We chose to focus on 2D dynamics in this work but the more complex 3D systems suggested by the reviewer are indeed interesting to study, and future work should consider studying them (we are not aware of any other work exploiting videos as raw measurements).

---

> > ### Comment · Reviewer_3pLg · 2021-09-01
> > **Review update**
> >
> > Thank you for your rebuttal. I will change my score to 6 as long as you promise to add a comprehensive description and discussion on the new 2D fluid dynamics example to the main paper. In particular, I think the new experiment should present not only the prediction error but also results about controlling the 2D fluids (similar to examples in [6]). Do you have such results, or did I miss them in your rebuttal?
> >
> > I also think it is necessary to comment on whether/how this approach applies to 3D examples. I am OK with leaving 3D examples as future work. Still, I request adding a few sentences in the conclusion section describing what you foresee are the potential technical challenges if people want to apply your approach to 3D.

---

> > > ### Author Response · Authors · 2021-09-01
> > > **After the initial response to reviewer 3pLg**
> > >
> > > We thank the reviewer for his/her comments. We commit of course to adding a comprehensive description and discussion on the new 2D fluid dynamics experiments to the main paper if it is accepted. We are currently working on including control in this setting, similar to [6], but conclusive results are not ready yet. The corresponding experiments will also be presented in the revision. At this point, 3D examples are left for future work, but we commit to discussing this issue in the revision as well.

---

### Official Review · Reviewer_1SAM · 2021-07-18

**Rating:** 6
**Confidence:** 4

**Summary:**

This paper, inspired by (approximations of) Koopman operator theory, proposes an adaptive algorithm that learns a (controlled) dynamics model online. The learned model consists of an encoder and estimated (and continuously updated) A, B matrices. The method is shown to work well (with low RMSE and good control performance) on several low-level control tasks with images as the observations (hence the encoder).

**Main Review:**

I think that this paper is interested and could be accepted for NeurIPS, however there are some improvements that could be done:

* Discussion of algorithmic limitations. Experiments could be extended to higher DOF control systems to address this issue. Failures/problems with auto-encoder should be mentioned, how/when do these effect the introduced approach? In 3.4 how would the approach work if one were learning and controlling at the same time? How would some of the hyperparameters influence the results if not specified well? e.g. h, \rho,

* How would the approach work if B(x) also depended on x, with x the image or if B(z), if z is the latent variable? In control theory, the control affine dynamics x_{t+1} = A(x) x_{t} + B(x) u_{t} is very popular.

* How does minimizing (12) work together with the least squares estimation for A, B? Maybe including algorithm (with the full loop) could clarify this.

* Koopman operator theory seems almost tangential to the introduced approach. How does this theory influence/guide the introduced approximate approach? When could the approximations be problematic, does the theory offer any insights?

* Comparisons: are there other control/model learning papers (also those that don't use autoencoders?) that work on images? Could these be included in the comparisons?

Minor comments:

- The title is too general, please change to restrict to the specific method that is introduced (adaptively learning latent linear models from images)

- There are a few mistakes, some hindering the readability, please go over the manuscript carefully.

- Captions are not adequate, the figures need to be self-explanatory. Check also line 251 for missing figure.

===== UPDATE =====

After reading the rebuttal and re-reading the paper, I still think that the paper can be accepted. The authors have replied well to most points raised also. However I cannot raise my score more because one of my main concerns, that of the Koopman operator theory not guiding the introduced methodology, was not answered sufficiently. For future work (regardless whether the paper is accepted or not), I would advise the authors to look at when or if Koopman operator theory could guide the design of approximate frameworks. When could features be sufficient for a control task? Can we know if by constraining the features and the dynamics model within some parametric form + dimensionality, we are not sacrificing much (i.e. staying 'close' to the operator P in some metric?)

**Time Spent Reviewing:**

10

---

> ### Author Response · Authors · 2021-08-10
> **Response to reviewer 1SAM**
>
> Thank you for your time spent reviewing and for your suggestions. We will try to address each one of your points in the following:
> > Discussion of algorithmic limitations. Experiments could be extended to higher DOF control systems to address this issue. Failures/problems with auto-encoder should be mentioned, how/when do these effect the introduced approach? In 3.4 how would the approach work if one were learning and controlling at the same time? How would some of the hyperparameters influence the results if not specified well? e.g. h, \rho,
> - We agree that our experiments should be extended to higher DOF control systems, and we have indeed extended them to computational fluid dynamics (CFD) data. We refer the reviewer to the common answer for more details.
> - The autoencoder architecture described in the submission works empirically well on the studied systems. Training it for reconstruction only (without prediction of future frames) yielded models with low RMSE and visually perfect reconstructions. Training it for both reconstruction (for the first m frames) and prediction(for the last $T-m$ frames) as in equation (12) was not problematic and did not decrease the quality of the reconstruction. However, on the more complex CFD data this simple architecture is not sufficient, in the sense that even without the prediction part, the reconstructions are noisy and blurry. In this case, we have turned to a slightly more complex architecture that includes one skip connection in one of the last levels of the autoencoder following the same strategy as in [26].
> For the purpose of control in the latent space, the reconstructions do not have to be perfect, what matters is the quality of the latent code, the decoder’s role being mostly to verify that enough information was captured.
> - We believe that including control in the learning framework will improve the approach as it will allow us to learn a representation explicitly suited to control. We are currently working on this extension.
> - As discussed in [19], the minimization problem we solve in equation (5) using the augmented Lagrangian in equation (6) converges for any value of $\rho > 0$. However, the convergence rate is proportional to $\rho^{-1}$: meaning that the problem will converge faster for small values of $\rho$, typically 1e-6. Also, too small or too large values of $\rho$ may decrease the numerical accuracy of the approach due to numerical rounding errors.
> $h$ was set to a higher value than 1 based on the intuition that at least two consecutive images should be considered to capture a system’s velocity. Higher values of $h$ led to higher-quality reconstructions but required more training data. In practice, $h$ was set to 2 for the simple and double pendulums as no significant improvement was achieved with higher values. On the cart pole system, increasing $h$ (up to 5) did improve the predictions quality.
> ___
> >How would the approach work if B(x) also depended on x, with x the image or if B(z), if z is the latent variable? In control theory, the control affine dynamics x_{t+1} = A(x) x_{t} + B(x) u_{t} is very popular.
> - In our approach, unless $A$ and $B$ are used right after being estimated or updated, $B$ is indeed independent of $x$ (or $z$). We followed previous work that also made the assumption of $B$ being independent of $x$ (or $z$) and were not limited by this assumption in the case of the systems we studied.
> In the case where this assumption does not hold and that we can not take advantage of the closed form solution allowed by LQR, we could consider iterative LQR [a], which will still be less expensive than model-based RL algorithms.
> We will add this discussion about the approximation of $B$ being independent of $x$ in the submission.
> ___
> >How does minimizing (12) work together with the least squares estimation for A, B? Maybe including algorithm (with the full loop) could clarify this.
> - For ease of reading, the estimation of $B$ doesn’t appear in Eq (12). If we were to add it, the norm in the second term of the loss would be $\|d_t^i - \Psi_\mu(A^{t-m}\Phi_\theta(d_m) + Bu_t)\|$.
> At each optimisation step, $A_i$ and $B_i$ are estimated using Eq (5) for each trajectory $i$ $(d_1 ^ i, d_T ^ i)$. They are estimated from the first m codes of $(d_t ^ i)_t$ obtained with the encoder $\Phi_\theta$.
> - $A_i$ and $B_i$  are then used to predict future codes through the relation $z_{t+1} ^ i  = A_i z_{t} ^i + B_i u_{t} ^ i $. The $(z_t^ i)_t$ are then decoded using $\Psi_\theta$, and parameters ($\theta$, $\mu$) of $\Phi_\theta$ and $\Psi_\mu$ are then updated.
> For the second term of eq (5) to be low, predictions have to be correct, thus $A_i$ and $B_i$ must be correct.
> We will clarify this in the revised version of the paper by adding the full algorithm.
> -  We will also include a figure that shows how the $\sum_{i,t} \|\|z_{t+1}^ i - A_i z_t ^i -  B_i u_t ^ i\|\|_2^2$ decreases during training without being explicitly minimized
> ___
> > Koopman operator theory seems almost tangential to the introduced approach. How does this theory influence/guide the introduced approximate approach? When could the approximations be problematic, does the theory offer any insights?
> -  Our approach is indeed guided by the Koopman operator theory, which guarantees the existence of a state-space representation where the dynamics of a studied system are linear. In our approach, we look for an approximation of the Perron-Frobenius operator (which acts on features of the observables of the system), adjoint to the Koopman operator (which acts directly on the observables of the system). We chose to look for this approximation using DMD, following previous approaches [6, 8, 12].
> In theory, the Koopman operator and the observable spaces are infinite-dimensional. In our approach, we look for finite-dimensional representations which dimension we explicitly choose. Doing so has not been a limitation in our experiments so far, however future work should consider better strategies to determine the latent dimension.
> The Koopman approximation could be problematic when applied to control as there is no theoretical ground for it. However, the approximation works well empirically in [6, 8, 16]. It was not a limitation in our experiments.
> ___
> > Comparisons: are there other control/model learning papers (also those that don't use autoencoders?) that work on images? Could these be included in the comparisons?
> - Reviewer 9oAb pointed us to Embed to Control, where the authors learn an embedding to a space where a locally linear model is built. However, several aspects of this method differ from ours. We list here the differences we also reported to reviewer 9oAb:
> [Watters]’s locally linear model (E2C) is not linear in the same sense as our model is, even locally. Indeed, from Eq. (10) of their paper, it appears that $A_t$ is a nonlinear function of $z_t$. From Eq. (8), we see that $z_{t+1}$ is sampled from a normal distribution whose mean is $A_t \mu_t + B_t u_t.$ Because $A_t$ is a nonlinear function of $z_t$, $z_{t+1}$ is then itself a nonlinear function of $z_t$. The only case in [Watters]’s approach where $z_{t+1}$ is a linear function of $z_t$ is in what they refer to as Global E2C.
> Also, [Watters] learns a state embedding which is system-specific, and a model of dynamics which is time-dependent. In contrast, our embedding is shared by physical systems of the same class with different physical parameters (masses, bar lengths, pole lengths), and our dynamics model is globally linear, as justified by Koopman theory (in a proper, usually infinite-dimensional state space, so our model is just an approximation of course, following DMD and EDMD). In control applications, the model of [Watters] is constructed from two video frames and updated at the control rate (typically a few milliseconds), whereas ours is constructed from as many as 50 frames, and updated at a much lower rate (typically every 1/60s when a new video frame is acquired). In practice, this allows us to predict sharp videos on much larger horizons than the ones considered in [Watters] (50 video frames instead of 10).
> Another minor difference with [Watters] is that we verify that the control sequence returned by solving equation (13) from our paper is efficient to control the actual system (by inputting the control sequence to the simulator and verifying that the system goes to the desired position), on top of being efficient to control the system in the latent space (by recursively doing $z_{t+1} = Az_t + B u_t$ with all controls from the sequence then decoding the sequence $[z_t]_t$.).
> Finally, although [Watters] states that its approach can easily be applied to globally linear settings, its experiments (Table 1 in [Watters]) show that, contrary to ours, the global version of their approach is inadequate in the case of the inverted pendulum.
> ___
> > Minor
> - We will remove “complex” from our title, and change “sensory input” to “image data” if CFD results are not added to the revised paper.
>
> [6] J. Morton, A. Jameson, M. J. Kochenderfer, and F. D. Witherden, “Deep dynamical modeling and control of unsteady fluid flows,” in NeurIPS 2018.
>
> [8] S. L. Brunton, B. W. Brunton, J. L. Proctor, and J. N. Kutz, “Koopman invariant subspaces and finite linear representations of nonlinear dynamical systems for control,” PLOS ONE, 2016.
>
> [12] N. Takeishi, Y. Kawahara, and T. Yairi, “Learning koopman invariant subspaces for dynamic mode decomposition,” in Advances in NeurIPS, 2017.
>
> [16] Y. Li, H. He, J. Wu, D. Katabi, and A. Torralba, “Learning compositional koopman operators for model-based control,” 2020
>
> [19] N. Parikh and S. Boyd, “Proximal algorithms,” 2014.
>
> [26] E. Denton and R. Fergus, “Stochastic video generation with a learned prior,” in ICML, PMLR, 2018
>
> [a] Li, Weiwei, and Emanuel Todorov. "Iterative linear quadratic regulator design for nonlinear biological movement systems." ICINCO (1). 2004.

---

> > ### Author Response · Authors · 2021-08-26
> > **After the initial response to reviewer 1SAM**
> >
> > We agree with the reviewer's comments: In fact, we believe that this is an issue with all DMD and EDMD approaches we are aware of: they are inspired by Koopman theory but typically do not give a true theoretical link between the theory and its practical implementation in finite-dimensional operators. DMD and EDMD can in fact be interepreted as least-squares approximations of the Koopman operator (or rather its adjoint, the Perron-Frobenius operator), but we are not aware of any theoretical bounds on the approximation, and believe that finding one remains an open problem. We have found a few papers discussing this issue:  in [b], Williams et al. show that EDMD modes are a good approximation of the true Koopman eigenfunctions for a specific problem when these are known analytically  (LTI system), and that in another special case where no analytical expression is available, but the dynamics are well understood (Duffing equation), the EDMD modes provide a correct parametrization of the system. Likewise, Zhang et al. show in [c] that, for some 2D nonlinear systems for which the true Koopman spectral decomposition is known, the DMD modes provide a good approximation of the corresponding Koopman eigenfunctions. Again, as far as we know, assessing the accuracy of the DMD/EDMD approximation to Koopman theory for nonlinear dynamic remains an open problem. If our paper is accepted, we will discuss this important issue and cite these two papers or any other references the reviewers would like to suggest.
> >
> > [b] Williams, M. O., Kevrekidis, I. G., & Rowley, C. W. (2015). A data–driven approximation of the koopman operator: Extending dynamic mode decomposition. Journal of Nonlinear Science, 25(6), 1307-1346.
> > [c] Zhang, H., Dawson, S., Rowley, C. W., Deem, E. A., & Cattafesta, L. N. (2017). Evaluating the accuracy of the dynamic mode decomposition. arXiv preprint arXiv:1710.00745.

---

### Official Review · Reviewer_9oAb · 2021-07-23

**Rating:** 6
**Confidence:** 5

**Summary:**

This work proposed an approach of online learning and control of nonlinear systems relying on the insights from Koopman operator theory, and included empirical examples.


**Ethical Concerns:**

The reviewer believes there is no significant ethical issue in this work.

**Limitations And Societal Impact:**

Limitations could be discussed more (for example, the concerns the reviewer mentioned above)
Social impact is discussed well.

**Main Review:**

The importance of online nonlinear control problem is clear and should be studied.
However, for this work, the reviewer has several major concerns.

Major concerns
1) The overall approach is very similar to embed to control [https://arxiv.org/pdf/1506.07365.pdf].
   Embed to control primarily considers locally linear model over latent space, but it also mentions that it can be easily applied to globally linear cases.
   It tackles the problem of controlling agents from raw image inputs and studies the empirical results thoroughly.
   Therefore, except for the fact that the current work explicitly says it is doing online updates, the reviewer hardly sees any difference from the line of work mentioned above.  (see [https://arxiv.org/pdf/1910.08264.pdf] as well)

2) For online update aspects; the difficulty of online learning for control problems basically stems from the difficulty of exploration.
   Without exploration, for example, [https://arxiv.org/pdf/1703.04680.pdf] this work studies convergence of EDMD under certain conditions, and it is straightforward to use this idea to study the trade-off between the amount of data and approximation accuracy, which can then be used for robust control for example.
   Also, [https://arxiv.org/pdf/1906.05194.pdf] this work already studies active learning of Koopman operator for optimal control problem under certain conditions, which is basically about the online learning.
   If the current work considers so carefully about the online learning problem either experimentally or theoretically that it gives us more insights than the line of work mentioned above, those should be clearly stated (the reviewer thinks it is not clarified in the current form)

3) The author(s) claim that this work is for controlling complex dynamical systems; however, as long as the underlying dynamics is pendulum, it would be hard to justify this claim.
   Also, it is not always possible to write a nonlinear controlled system by  z_t+1 = Az_t + Bu+t even if the space of z is carefully selected.  (B should be dependent on x in general)
   The advantage of this linear model is the ease of computing optimal control using Riccati equation; however, if this transformation cannot be applied to general nonlinear systems, it is hard to say that this formulation can be applied to complex dynamical systems.  Then, what are the advantages of this approach over existing model-based RL (which includes online algorithm as well)?

Here are some minor comments
1) line 251;  what is "??" here?  there are some minor typos, please check them again.

-------------
review updated after the initial response

**Time Spent Reviewing:**

4 hours

---

> ### Author Response · Authors · 2021-08-10
> **Response to reviewer 9oAb**
>
> Thank you for your comments, and for pointing us to references we didn’t know. We will try to address each one of your concerns in the following:
>
> > The overall approach is very similar to embed to control [https://arxiv.org/pdf/1506.07365.pdf]. Embed to control primarily considers locally linear model over latent space, but it also mentions that it can be easily applied to globally linear cases. It tackles the problem of controlling agents from raw image inputs and studies the empirical results thoroughly. Therefore, except for the fact that the current work explicitly says it is doing online updates, the reviewer hardly sees any difference from the line of work mentioned above. (see [https://arxiv.org/pdf/1910.08264.pdf] as well)
> - We agree with the reviewer that both of these publications (which we will refer to from now on as [Watters] and [Li] from the names of their first authors), are relevant, and we will cite and discuss them in the revision of our paper if it is accepted. Like us, [Watters] and [Li] seek state space representation where the dynamics are linear. There are, however, several major differences between our approaches.
> - [Watters]’s locally linear model (E2C) is not linear in the same sense as our model is, even locally. Indeed, from Eq. (10) of their paper, the update matrix $A_t$ itself is a nonlinear function of $z_t$ (the outout of a neural network). From Eq. (8), we see that $z_{t+1}$ is sampled from a normal distribution whose mean is $A_t \mu_t + B_t u_t$. Because $A_t$ is a nonlinear function of $z_t$, $z_{t+1}$ is then itself a nonlinear function of $z_t$. The only case in [Watters]’s approach where $z_{t+1}$ is a linear function of $z_t$ is in what they refer to as Global E2C.
> Also, [Watters] learns a state embedding which is system-specific, and a model of dynamics which is time-dependent. In contrast, our embedding is shared by physical systems of the same class with different physical parameters (masses, bar lengths, pole lengths), and our dynamics model is globally linear, as justified by Koopman theory (in a proper, usually infinite-dimensional state space, so our model is just an approximation of course, following DMD and EDMD). In control applications, the model of [Watters] is constructed from two video frames and updated at the control rate (typically a few milliseconds), whereas ours is constructed from as many as 50 frames, and updated at a much lower rate (typically every 1/60s when a new video frame is acquired). In practice, this allows us to predict sharp videos on much longer horizons than the ones considered in [Watters] (50 video frames instead of 10). Another minor difference with [Watters] is that we verify that the control sequence returned by solving equation (13) from our paper is efficient to control the actual system (by inputting the control sequence to the simulator and verifying that the system goes to the desired position), on top of being efficient to control the system in the latent space (by recursively applying $z_{t+1} = Az_t + B u_t$ with all controls from the sequence then decoding the sequence $[z_t]_t$.).
> Finally, although [Watters] states that its approach can easily be applied to globally linear settings, its experiments (Table 1 in [Watters]) show that, contrary to ours, the global version of their approach does not work well for the inverted pendulum.
> - Like us, [Li] learns a state representation using Koopman theory to find a linear representation of the dynamics. However, unlike us, it assumes that the states themselves are available during training. In contrast, we assume no knowledge of the states and construct their representation directly from sensory data, a video stream in our case.
>
> ___
>
> > For online update aspects; the difficulty of online learning for control problems basically stems from the difficulty of exploration. Without exploration, for example, [https://arxiv.org/pdf/1703.04680.pdf] this work studies convergence of EDMD under certain conditions, and it is straightforward to use this idea to study the trade-off between the amount of data and approximation accuracy, which can then be used for robust control for example. Also, [https://arxiv.org/pdf/1906.05194.pdf] this work already studies active learning of Koopman operator for optimal control problem under certain conditions, which is basically about the online learning. If the current work considers so carefully about the online learning problem either experimentally or theoretically that it gives us more insights than the line of work mentioned above, those should be clearly stated (the reviewer thinks it is not clarified in the current form)
>
> - We agree that exploration is an issue for online learning. To deal with this, we have chosen to handle it in our data generation  process where we generate controlled trajectories to match trajectories that largely cover the working space. Yet, our ability to correctly capture the system dynamics is only guaranteed over the training distribution. As future work, we plan to extend our approach to account for active learning to also consider system configurations and control not covered by the initial training sets. This will be discussed in the revision.
> - There are several differences between the work of Abraham et al. and ours.
> First, Abraham et al. try to find a state-space representation where a global Koopman operator can be approximated for a given system from multiple trajectories. In our case, a different Koopman operator approximation computed from the measurements is associated with each trajectory. This allows us to capture the evolution of multiple dynamical systems with different physical parameters. This also implies that our learning strategy is different from that of Abraham et al. While their training dataset is made of pairs of consecutive states all obtained from executions of the same system (and eventually the control input that allowed the transition from one to the next), ours is composed of longer measurement trajectories (in image space) resulting from execution of different systems.
> Second, Abraham et al. assume the state of the system is known. They either handcraft observable functions on the states using prior knowledge about the studied system or learn parameters of a representation that also includes the state of the system. In our approach, we assume we do not have access to the full state of the system, which is the case in most practical setups (fluid mechanics, complex contact interactions in robotics, etc.), and we do not include any prior knowledge about the system when learning its representation. All we assume known are video measurements of it.
> - We will include the comparisons to this approach in the revised version of our paper.
>
> ___
> > The author(s) claim that this work is for controlling complex dynamical systems; however, as long as the underlying dynamics is pendulum, it would be hard to justify this claim. Also, it is not always possible to write a nonlinear controlled system by z_t+1 = Az_t + Bu+t even if the space of z is carefully selected. (B should be dependent on x in general) The advantage of this linear model is the ease of computing optimal control using Riccati equation; however, if this transformation cannot be applied to general nonlinear systems, it is hard to say that this formulation can be applied to complex dynamical systems. Then, what are the advantages of this approach over existing model-based RL (which includes online algorithm as well)?
> - We will tone down our claims of handling complex dynamical systems. Note, however, that the cart pole and double pendulum systems are chaotic systems with dynamics that are hard to capture from raw images.
> We have also recently applied our approach to a more complex dataset from computational fluid dynamics. We refer the reviewer to the common answer for more details.
> In our approach, unless A and B are used right after being estimated or updated, $B$ is indeed independent of $x$ (or $z$). We have followed previous work [6, 16] that also makes this assumption and have not found it to pose problems in the case of the systems we studied.
> - In the case where this assumption does not hold and that we can not take advantage of the closed form solution allowed by LQR, we could consider iterative LQR [a], which will still be less expensive than model-based RL algorithms. We will add this discussion about the approximation of $B$ being independent of $x$ in the final version of the paper.
>
> [6] J. Morton, A. Jameson, M. J. Kochenderfer, and F. D. Witherden, “Deep dynamical modeling and control of unsteady fluid flows,” in Advances in Neural Information Processing Systems 31: Annual Conference on Neural Information Processing Systems 2018, NeurIPS 2018, 2018.
>
> [16] Y. Li, H. He, J. Wu, D. Katabi, and A. Torralba, “Learning compositional koopman operators for model-based control,” 2020
>
> [a] Li, Weiwei, and Emanuel Todorov. "Iterative linear quadratic regulator design for nonlinear biological movement systems." ICINCO (1). 2004.

---

> > ### Comment · Reviewer_9oAb · 2021-08-19
> > **After the initial response**
> >
> > Thank you for the response;
> >
> > Regarding the author response "Finally, although [Watters] states that its approach can easily be applied to globally linear settings, its experiments (Table 1 in [Watters]) show that, contrary to ours, the global version of their approach does not work well for the inverted pendulum.", the reviewer thinks this seems to be the only main difference after all since the work [Watters] anyway considers globally linear model.
> >
> > Regarding the response "Abraham et al. assume the state of the system is known. They either handcraft observable functions on the states using prior knowledge about the studied system or learn parameters of a representation that also includes the state of the system"; it seems this is because the work [Abraham] considers optimal control problem at the end and it requires to embrace the state in the observable.  Otherwise, the same framework should work without the raw state information.
> >
> > However, considering this is a conference, given that the author(s) will provide clear and careful comparisons to the existing work in the final version and focuses more on the novel aspect of the proposed approach, the reviewer raises the score to 6 from 4.

---

> > > ### Author Response · Authors · 2021-08-24
> > > **After the initial response to reviewer 9oAb**
> > >
> > > Thank you for your response. We commit of course to following your suggestions in the revised version of the paper if it is accepted.

---

### Author Response · Authors · 2021-08-10
**Response to all the reviewers**

We would like to thank all the reviewers for the time spent reviewing our submission, and for their helpful remarks and questions. We have tried to address the concerns of each reviewer in our individual responses, and include below a summary of these responses.
- We will tone down the writing in our submission, fix typos, and make figures more self-explanatory. We apologize for the missing figure reference in line 251 which should have been Fig. 3.
- To show its effectiveness in more complex settings, we have also applied our approach to computational fluid dynamics data generated following the protocol described in [6], demonstrating the wider applicability of our approach. We report in the following table the mean L1-norm of the prediction error on 4 quantities over 64 time steps (Density, x-momentum, y-momentum, and energy) which were all rescaled between 0 and 1.

|Time step | 10 | 20 | 30 | 40 | 50 | 60 |
|--- | --- | --- | --- | --- | ---| --- |
|Without updates | 2.27e-3 | 2.29e-3 | 2.31e-3 | 2.43e-3 | 2.51e-3| 2.74e-3|
|With one update at step 20 | 2.02e-3  | 2.06e-3 | 2.06e-3 | 2.09e-3 | 2.18e-3  | 2.33e-3 |
|With updates at every step | 1.93e-3 | 1.93e-3 | 1.93e-3 | 1.96e-3 | 1.96e-3 | 2.01e-3 |


Our results are of the same magnitude as the results reported in [6]. Note that the prediction error grows a little slower with time when online updates are performed.


- In this work, we focus on learning from raw video recordings an embedding of the (unknown) states of a dynamical system in a latent space where the evolution of the system can be approximated with a linear model. We aim to use this learned embedding to control the system by performing LQR in the latent space.
Our approach includes two phases. During the first phase, we learn an embedding (through an encoder) that is common to various systems of the same family (e.g., cartpole systems with different masses, different pole lengths and different cart lengths) from which a linear model can be extracted and used for prediction of future latent codes. We verify that the estimated model is correct by comparing its predictions to observed videos.
To take into consideration the fact that the dynamics of complex dynamical systems cannot usually be fully captured by a linear model estimated from an initial trajectory, we force our representation to be adaptable using online learning. Given an initial trajectory of a system, a linear model (A and B matrices) is estimated in the learned latent space and updated with new measurements (a few time steps after the model estimation). Performing these online updates result in learned representations that are adaptable to new measurements, and help avoid diverging predictions even for chaotic systems (cartpole).
During the second phase, the aim is to control a system to take it from an initial position to a target position, both specified in the measurement space (they are both images). We map an observed trajectory of this system using the learned encoder from the first phase to the learned latent space, and use this embedding of the trajectory to estimate the A and B matrices associated with this system. They are then plugged to solve an LQR problem associated with the task of taking the system from initial to target position, which returns a sequence of control inputs to achieve this task. Figure 6 from the main submission and Fig. 7 from the supplementary material show examples of controlled systems where the controls were applied in the latent space (the resulting video sequence was then obtained using the decoder learned during the first phase).
___

[6] J. Morton, A. Jameson, M. J. Kochenderfer, and F. D. Witherden, “Deep dynamical modeling and control of unsteady fluid flows,” in NeurIPS 2018.

---

### Decision · Program_Chairs · 2021-09-27

**Decision:**

Accept (Poster)

**Comment:**

From the SAC. This is an instance where the rebuttal and the discussion worked. While the original decision for this paper was to not accept, it is being raised to a recommended accept. The primary reason is the quality of the rebuttal, and the useful technical discussion between authors and reviewers that ensued that seems to have been revealing. To the authors: I trust that you will take all reviewer feedback into account and most importantly, that all of the things in your rebuttal and discussion that were promised will be done in the next version of the paper.